# Relationships between Recent Suicidal Ideation and Recent, State, Trait and Musical Anhedonias in Depression

**DOI:** 10.3390/ijerph192316147

**Published:** 2022-12-02

**Authors:** Matthieu Hein, François-Xavier Dekeuleneer, Olivier Hennebert, Dephine Skrjanc, Emilie Oudart, Anaïs Mungo, Marianne Rotsaert, Gwenolé Loas

**Affiliations:** 1Department of Psychiatry, Laboratory of Psychiatric Research (ULB 266), Cliniques Universitaires de Bruxelles, Université Libre de Bruxelles (ULB), 1070 Bruxelles, Belgium; 2Department of Child Psychiatry, Laboratory of Psychiatric Research (ULB 266), Cliniques Universitaires de Bruxelles, Université Libre de Bruxelles (ULB), 1070 Bruxelles, Belgium; 3Department of Psychology, Université Libre de Bruxelles (ULB), 1070 Bruxelles, Belgium

**Keywords:** anhedonia, state anhedonia, trait anhedonia, musical anhedonia, depression, melancholia, suicidal ideation, suicide

## Abstract

The aim of the study was to explore in depression the relationship between recent suicidal ideation and the different anhedonias taking into account the severity of depression. Recent studies have suggested that recent change of anhedonia and not state or trait anhedonia is associated with recent suicidal ideations even when the level of depression is controlled. Three samples were used (74 severe major depressives, 43 outpatients with somatic disorders presenting mild or moderate depression and 36 mild or moderate depressives hospitalized in the intensive coronary unit). Recent change of anhedonia was rated by the anhedonia subscale of the Beck Depression Inventory (BDI-II), state anhedonia by the Snaith–Hamilton Pleasure Scale (SHAPS), trait anhedonia by the TEPS (Temporal Experience of Pleasure Scale), musical anhedonia by the BMRQ (Barcelona Music Reward Questionnaire), social recent change of anhedonia by the SLIPS (Specific Loss of Interest and Pleasure Scale), the severity of depression by the BDI-II and the distinction between melancholic and non-melancholic was found using a subscale of the BDI-II. Bivariate and multivariate regression analyses were performed in each sample. In severe major depressives and, notably, in melancholia, recent suicidal ideation was associated with trait anhedonia; however, in mild or moderate depression, recent suicidal ideation was associated with recent change of anhedonia. Musical anhedonia and social recent change of anhedonia were not associated with recent suicidal ideation. Trait anhedonia could be, in severe depression, a strong predictor of recent suicidal ideation.

## 1. Introduction

Anhedonia is defined as the inability to experience pleasure [1]. A scientific study on hedonia has shown that reward comprises several components and processes of wanting, liking and learning that are related to the appetitive, consummatory and satiety phases of the pleasure cycle [2]. In psychopathology, anhedonia is considered a multi-faceted construct, including trait level, state level and recent change [3]. Among these three facets, physical–sensorial and social anhedonias can be distinguished that correspond to the liking and wanting components of pleasure.

Several recent studies including a meta-analysis [4] have reported that anhedonia has a strong relationship with suicidal ideation independently of depression. Several authors have suggested to take into account the heterogeneity of anhedonia when examining its relationship with suicidal ideation. One hypothesis is that state anhedonia and not trait anhedonia is associated with suicidal ideations [5], and another hypothesis is that recent change of state anhedonia and not trait anhedonia is associated with suicidal ideations [3].

According to Winer et al. [3], the available tools to measure state anhedonia, such as the Snaith–Hamilton Pleasure Scale (SHAPS, [6]), do not allow for a clear distinction with trait anhedonia. Several studies in university and undergraduate students [7,8,9,10] as well as on clinical samples [3,11,12] measuring state anhedonia with the SHAPS, recent change of state anhedonia with the Specific Loss of Interest and Pleasure Scale (SLIPS, [13]) or the anhedonia subscale of the Beck Depression Inventory (BDI-II, [14]), and trait anhedonia with the Temporal Experience of Pleasure Scale (TEPS, [15]), reported that recent change of anhedonia was associated with suicidal ideations. State anhedonia and trait anhedonia, however, were not.

In mood disorders, including subjects with anxiety, depressive or bipolar disorders, six studies have reported contradictory results concerning the relationships between suicidal ideation and recent change, state or trait anhedonia [10,11,16,17,18,19].

To explain the discrepancy between the results of the six studies, several explanations can be proposed. First, the samples were heterogeneous and did not exclusively include mood disorders [10,16]. Second, mood disorders included not only depressive disorders but also anxiety disorders [11]; the relationship between anhedonia and suicidal ideations could be different between anxiety and depressive disorders. Third, the severity of depression could explain the different relationships between anhedonia and suicidal ideation. For example, severe depression and, notably, melancholia are characterized by a high physical trait-anhedonia level compared to non-melancholia [20].

Thus, it could be interesting to explore the relationships between suicidal ideation and the different anhedonias only in depression taking into account the severity of depression and the distinction between melancholia and non-melancholia.

However, Winer et al. [3] have suggested that the social component of recent change of anhedonia, as rated by the 12th item « Loss of interest » of the BDI-II or by the SLIPS, was the most predictive component of suicidal ideation. This suggestion has been confirmed in several studies that included high school students, university students and psychiatric patients [3,7,8,10,21,22]. Two studies on physicians and medical students did not report such an association [9,23].

In depression, the hypothesis has not been tested as two studies explored either psychiatric patients [3] or a mixed sample of healthy subjects and major depressives [21].

Moreover, to the best of our knowledge, the relationship between musical anhedonia and suicidal ideation has not been explored [24]. Most people enjoy music but around 5% do not [25], and this feature is called musical anhedonia. There is a controversy over whether a lack of musical enjoyment is a specific type of anhedonia or part of the general anhedonia construct. A recent study [26] on 153 music professionals and 303 students showed that musical anhedonia was strongly related to social anhedonia.

A review of the literature shows that three points deserve to be clarified concerning the links between the different anhedonias and suicidal thoughts. First, the relationships between suicidal ideations and the different anhedonias specifically in depression, taking into account the severity and the melancholia versus non-melancholia distinction, have not been explored. Second, the particular role of the social aspect of anhedonia has not been explored specifically in depression. Third, to the best of our knowledge, the relationship between musical anhedonia and suicidal ideation has never been explored.

Thus, the aim of the present study was to explore the relationships between recent suicidal ideation and recent change, state, trait and musical anhedonia in depressive disorders taking into account the depressive level, the subtype of melancholia and the social component of recent change of anhedonia.

## 2. Materials and Methods

### 2.1. Participants 

In order to examine the relationships between anhedonias and recent suicidal ideation, three samples were recruited (see Table 1).

Sample 1: A total of 96 inpatients or outpatients with a DSM-5 diagnosis of major depressive disorder and a score of at least 12 on the Beck Depression Inventory (BDI-II) were included in the study. There were 30 men and 66 women with a mean age of 44.73 years (SD = 12.65, range: 20–83). The subjects were recruited from psychiatric departments (Erasme hospital of Brussels and CHU of Tivoli, Belgium). All the subjects received antidepressants. The aim of the study was explained in an information document, and all participants gave their signed written informed consent. The study was approved by the Ethics Committee of the Hôpital Erasme (P2017/080 & P2017/216). A total of 74 subjects had severe depression with a score of ≥28 on the BDI-II. Pizzagalli et al. [27] have proposed for the diagnosis of melancholia a subscale of the BDI-II summing six items: #4 « loss of pleasure »; #5 « guilty feelings »; #11 « agitation »; #12 « loss of interest »; #16b « early morning awakening »; and #21 « loss of interest in sex ». The 74 subjects were divided into 2 subgroups using a cutoff score ≥ the mean of the subscale more than one standard deviation (58 non-melancholics and 16 melancholics). Taking into account that three of the six items of the melancholia subscale rate anhedonia, the score without these three items was calculated. Using a cutoff score ≥ the mean of the subscale more than one standard deviation, there were 58 non-melancholics and 16 melancholics. For the two classifications, the overall corrected fraction was 84% with a Kappa coefficient of 0.58. We reported only the results for the three-item subscale.

Sample 2: A total of 94 outpatients consulting with general practitioners for various disorders were included in the second sample. There were 60 men and 34 women with a mean age of 41.16 years (SD = 18.17; range: 18–79). The aim of the study was explained in an information document, and all participants gave their signed written informed consent. The study was approved by the Ethics Committee of the Hôpital Erasme (P2017/218). Using the BDI-II, 43 subjects were depressed (BDI-II ≥ 12), and 51 were not (BDI-II < 12).

Sample 3: A total of 100 subjects (66 men and 34 women) with a mean age of 65.9 years (SD = 14.04; range: 22–90) were recruited from the intensive coronary unit and the cardiology department of the Erasme hospital. The diagnoses were 61 angors or myocardial infarctions, 24 heart failures, 9 arrythmias, 3 valvulopathies and 3 heart transplants. The study was approved by the Ethics Committee of the Hôpital Erasme (P2014/260). The aim of the study was explained in an information document, and all participants gave their signed written informed consent. Among the 100 subjects, 36 had a score ≥ 12 on the BDI-II.

### 2.2. Measures 

Self-rating scales were used exclusively to evaluate recent suicidal ideation, depression and anhedonia.

Suicidal ideation (SID) was rated using the “Suicidal thoughts and wishes” item of the BDI-II in which 0 is “I don’t have any thoughts of killing myself” and 3 is “I would like to kill myself if I had the chance”.

Depression was rated using the revised version of the Beck Depression Inventory (BDI-II). The BDI uses statements that best describe how the individual has felt during the previous two weeks. The French version of the BDI-II has satisfactory psychometric properties [28]. The total score ranges from 0 to 63, and higher total scores indicate more severe depressive symptoms. The cutoff scores for the French version are ≥12 for mild depression and ≥28 for severe depression. For the present study and only in sample 1, we used the melancholia subscale proposed by Pizagalli et al. [27] minus the three items rating anhedonia. Moreover, for the present study and to control depression in the regression analyses, cognitive–affective symptoms of depression (CA-BDI) were assessed using the items relating to past failure, guilty feelings, punishment feelings, self-dislike, self-criticalness and worthlessness. This subscale of the BDI has been used previously by Winer et al. [3].

#### 2.2.1. Anhedonia

Several measures of anhedonia were used to rate trait anhedonia, state anhedonia, recent change of state anhedonia and musical anhedonia.

Trait anhedonia was rated using two rating scales. The PAS anticipatory (PAS-ANT) and PAS consummatory (PAS-CONS) scales have been designed by taking into account correlations between the items of the revised Physical Anhedonia Scale (PAS) and the TEPS-CONS or TEPS-ANT scales [29]. Ten items from the PAS-ANT and 16 items from the PAS-CONS subscales were selected. These two subscales presented satisfactory psychometric properties [29]. Total scores for the PAS-ANT and PAS-CONS range from 0 to 10 and 0 to 16, respectively, and are related to the level of trait anhedonia. The Temporal Experience of Pleasure Scale (TEPS, [15]) is a short scale that evaluates the existence of two different subcomponents of the chronic hedonic experience, such as the anticipatory and consummatory. The anticipatory pleasure is characterized by the individual expectation of receiving a pleasurable reward, and the consummatory pleasure accounts for the feeling of satisfaction in response to the reward. The TEPS consists of 18 items, each of which is rated from 1 (very false for me) to 6 (very true for me). Total TEPS scores range from 18 to 108 with a lower score reflecting a greater severity of anhedonia. TEPS anticipatory (TEPS-ANT) and consummatory (TEPS-CONS) subscales contain, respectively, 10 and 8 items. The French version of the TEPS had satisfactory psychometric properties [29].

State anhedonia was rated using the SHAPS (Snaith et al., [6]) that evaluates an individual’s state of pleasure experienced in recent days (for example, “I have enjoyed being with my family or close friends”). The scale consists of 14 items rated on a 4-point Likert scale from 1 (“I strongly agree”) to 4 (“I strongly disagree”). The total score ranges from 14 to 56. The level of pleasure is inversely related to the score of the scale; therefore, a higher score reflects a greater severity of anhedonia. The French version of the scale has satisfactory psychometric properties [30] with a value of 0.8 for the Cronbach alpha and a test-retest r of 0.56 over a one-month period.

Recent change of anhedonia was evaluated using two rating scales. The anhedonia subscale (ANH-BDI) of the Beck Depression Inventory (BDI-II, [14]) that contains three items (Item 4 or Loss of Pleasure (LP) ‘I can’t get any pleasure from the things I used to enjoy’; item 12 or Loss of Interest (LI) ‘It’s hard to get interested in anything’; and item 21 or Loss of Interest in Sex (LIS) ‘I have lost interest in sex completely’). Higher scores indicate more severe anhedonia. This subscale has been validated [31] in several samples of psychiatric subjects with the Cronbach alpha ranging from 0.57 to 0.73. Moreover, one confirmatory factorial analysis found that the two-factor model of the BDI-II (anhedonia-subscale and BDI with the remaining 18 items) had higher adequacy indices than the one-factor model [31]. The Specific Loss of Interest and Pleasure Scale (SLIPS, [13]) is a 23-question self-report measure that asks about changes in the ability to get interested or take pleasure in, primarily, social experiences. The SLIPS is scored on a four-point scale (0–3) resulting in a score range of 0–69 with a higher score reflecting a greater severity of anhedonia. The French version of the SLIPS had satisfactory psychometric properties [32]. The social component of anhedonia is measured either by item # 12 of the BDI-II (LI-BDI) or by the SLIPS.

#### 2.2.2. Musical Anhedonia

The Barcelona Music Reward Questionnaire (BMRQ, [33]) is a 20-item questionnaire designed to measure musical-reward experiences as a combination of five factors: musical seeking (MS), emotion evocation (EE), mood regulation (MR), sensory–motor (SM), and social reward (SR). A global score (music reward, MuR) is also available. Lower scores on the different factors or on the global score reflect a greater severity of anhedonia. The French version of the BMRQ has satisfactory psychometric properties [34].

The subjects of the three samples filled out the BDI-II and the SHAPS. The TEPS and the SLIPS were filled out only by the subjects of sample 1, the PAS were filled out by the subjects of samples 1 and 3 and the BMRQ was filled out by the subjects of samples 1 and 2.

### 2.3. Statistical Analyses

The analyses were performed for sample 1 on the 74 severe major depressives (score BDI-II ≥ 28) and on the 16 melancholics; the analyses were performed for samples 2 and 3 on the 43 and 36 depressives (score BDI-II ≥ 12), respectively.

In each sample, bivariate analyses and then multivariate analyses were performed.

First, bivariate statistical analyses were performed using the dependent variable (Suicidal ideation, SID) and the independent variables (ANH-BDI, LI-BDI, SHAPS, SLIPS, PAS-ANT, PAS-CONS, TEPS-ANT, TEPS-CONS, CA-BDI, MS, EE, MR, SM, SR, MuR, group (melancholia versus non-melancholia) and two covariates (gender, age)). The Pearson correlation and Student’s t-test were performed. The level of significance was *p* ≤ 0.05.

Second, multiple regressions were performed using recent change of suicidal ideation as dependent variables, covariates and independent variables, which had been found to be significant in the bivariate analyses. Ridge regressions were used to take into account the multicollinearity between the variables. Since suicidal ideation is an ordinal variable, a general nonlinear–linear regression model (linear ordinal multinomial regression model with logistic function) was also used.

Only in the first sample, bivariate and multivariate analyses were performed on the melancholic group (N = 16).

## 3. Results

First sample (74 major depressives with BDI-II ≥ 28).

The suicidal ideation item of the BDI-II correlated significantly with TEPS-CONS (r= −0.33, *p* < 0.05), PAS-ANT (r= 0.4, *p* <0.05), PAS-CONS (r = 0.35, *p* < 0.05), SHAPS (r = 0.3, *p* < 0.05) and CA-BDI (r = 0.28, *p* < 0.05).

Two multiple regressions were performed. The first with TEPS-CONS, SHAPS and CA-BDI as predictors and the second with PAS-ANT, PAS-CONS, SHAPS and CA-BDI as predictors. For the first regression, the result was significant (F (3, 70) = 4.24, *p* = 0.0082), and TEPS-CONS was a significant predictor (t = 2.43 *p* = 0.018). The corresponding linear ordinal multinomial regression reported close results with CA-BDI as a significant predictor (Wald = 4.24, *p* = 0.04) and a Wald value close to significance for TEPS-CONS (Wald = 3.67, *p* = 0.055). For the second regression, the result was significant (F (4, 69) = 5.02, *p* = 0.0013) with two significant predictors, PAS-ANT (t = 2.4 *p* = 0.019) and CA-BDI (t = 2.04, *p* = 0.044). For the corresponding linear ordinal multinomial regression, the Wald values of PAS-ANT and CA-BDI were, respectively, 6.07 (*p* = 0.014) and 3.9 (*p* = 0.048). On the melancholic group, the suicidal ideation item of the BDI-II correlated significantly with TEPS-CONS (r = −0.51, *p* < 0.05) and CA-BDI (r = 0.6, *p* < 0.05). The multiple regression was significant (F (2, 13) = 4.31, *p* = 0.037) without significant predictors.

Second sample (the analyses were performed on the 43 subjects having moderate depression (BDI ≥ 12)).

The suicidal ideation item of the BDI-II correlated significantly with ANH-BDI (r= −0.37, *p* < 0.05) and MS (r= −0.31, *p* <0.05). The multiple regression was significant (F (2, 40) = 4.37, *p* = 0.019). Among the predictors, only ANH-BDI was significant (t = 2.15, *p* = 0.038). For the corresponding linear ordinal multinomial regression, the Wald values of ANH-BDI and MS were, respectively, 4.22 (*p* = 0.04) and 1.56 (*p* = 0.21).

Third sample (the analyses were performed on the 36 subjects who had a score ≥ 12 on the BDI).

The suicidal ideation item of the BDI-II correlated significantly with CA-BDI (r = 0.4, *p* < 0.05).

## 4. Discussion

The present study had the aim to explore the relationships between the different anhedonias and recent suicidal ideations in depressive subjects taking into account the severity of the depression. In three different samples of depressive subjects presenting either mild, moderate or severe depression, we reported six main results (four positive and two negative).

### 4.1. Concerning Major Depressives

First, recent suicidal ideations were associated with trait anhedonia in severe depression, notably in melancholia. State anhedonia was also associated with recent suicidal ideations in severe depression. Several studies have explored the relationships between suicidal ideations and trait or state anhedonia in psychiatric samples including, notably, major depression.

Two studies reported contradictory results concerning the relationship between suicidal ideations and trait anhedonia in major depressives.

Sagud et al. [19] explored the association of physical and social anhedonia with recent suicidal ideation in three groups of subjects: non-psychiatric controls (N = 193), schizophrenia (N = 312) and major depressive disorder (N = 178). Recent suicidal ideation was rated using item 10 of the MADRS and physical and social anhedonia using the Chapman revised social and anhedonia scales. Only in major depressive disorders and not in schizophrenia, recent suicidal ideation was associated with physical and social anhedonia. Yang et al. [10] investigated whether state, trait and recent change in anhedonia predicted recent suicidal ideation in a sample of 859 subjects including 448 major depressives. Logistic regressions in the full sample as well as in the depressive sample found that only recent change of anhedonia, rated by the anhedonia subscale of the BDI-II, was a significant predictor of recent suicidal ideation and that state anhedonia, rated by the SHAPS, and trait anhedonia, rated by the TEPS, were not.

In these two studies, the severity of depression was lower than those of the present study (median score of 20 on the MADRS for Sagud et al. [19] and mean score lower than 28 on the BDI for Yang et al. [10]).

Second, the relationship between suicidal ideations and state anhedonia in major depression had been reported in three studies.

Ballard et al. [16] included 100 participants with treatment-resistant major depressive or bipolar disorders without psychotic features. Suicidal ideation was rated using the Beck Scale for Suicide Ideation and, notably, the first five items were included in the analyses. State anhedonia was rated using the SHAPS and recent change of anhedonia using the anhedonia subscale of the BDI-II. Depression was rated using the Hamilton Depression Rating Scale. Hierarchical linear regression found that state anhedonia and not recent change of anhedonia was a significant predictor of suicidal ideation even when depressive symptoms were controlled.

Hawes et al. [17] explored on 395 psychiatric outpatients, including 67% of mood disorders (174 depressive disorders, 44 anxiety disorders and 46 bipolar disorders), the relationships between trait or state anhedonia rated by a modified version of the SHAPS and suicidal ideation rated by the Beck Scale for Suicide Ideation. The subjects were divided into three groups (non-anhedonic, trait-anhedonic and state-anhedonic) and followed during one month. Controlled for symptoms of depression and anxiety, state anhedonia was cross-sectionally and prospectively associated with greater severity of suicidal ideation compared to the non-anhedonic group and the trait-anhedonic group.

Ducasse et al. [18] explored the association between anhedonia and suicidal ideation. A total of 2839 outpatients with mood disorders were recruited and followed during 3 years. State anhedonia was rated using the SHAPS and item 13 of the Quick Inventory of Depressive Symptomatology Scale. Patients with mood disorders and anhedonia at least at one follow-up visit had a 1.4-fold higher risk of suicidal ideation even after adjustment for confounding factors of suicide risk (for example, depressive severity).

### 4.2. Concerning Mild or Moderate Depressions

First, recent suicidal ideations were associated with recent change of anhedonia in mild or moderate depression but not in severe depression and, notably, melancholia.

Several studies have reported significant associations between recent suicidal ideations and recent change of anhedonia in psychiatric samples including depression.

In a six-week follow-up of 1529 adult psychiatric inpatients with at least 50% having mood or anxiety disorders, Winer et al. [3] reported significant associations at admission and at termination between recent suicidal ideations rated by the “Suicidal thoughts and wishes” item of the BDI-II and anhedonia rated by the ANH-BDI.

Loas et al. [11] found in a sample of 122 patients with mood or anxiety disorders, including two bipolar and 35 unipolar depressions, that recent suicidal ideation rated by the 9th item of the Beck Depression Inventory-II was significantly associated with recent change of anhedonia rated by the anhedonia subscale of the BDI-II and with the consummatory subscale of the physical anhedonia scale rating trait anhedonia. In this study, the correlations between recent suicidal ideation and the consummatory or anticipatory subscale of the Temporal Experience of Pleasure scale (TEPS) rating trait anhedonia were not significant.

### 4.3. Concerning Mild, Moderate or Severe Depressions

First, recent suicidal ideations were associated with depression severity in two of the three groups of depressive subjects independently of anhedonia as shown by the significant association between recent suicidal ideations and score on the cognitive–affective symptoms of depression (CA-BDI). The results of the present study partly confirmed the significant relationship between suicidal ideation and depression intensity even when anhedonia was not taken into account in the measure of depression [4].

Second, recent suicidal ideations were not associated with musical anhedonia in depression. To our knowledge, no other study has explored this relationship.

Third, recent suicidal ideation and social anhedonia rated by item 12 of the BDI-II or by the SLIPS were not associated in depression. Among the eight studies, none explored this relationship only in depression.

Fourth, higher scores of suicidal ideations were reported in severe major depressives than in the two other groups. A high prevalence of suicidal ideation has been reported among individuals with severe depression and seems to be one of the preconditions for suicide attempts [35].

## 5. Conclusions

The present study has several limitations. The study includes only a cross-sectional design, and the direction of causality among the assessed variables cannot be reported. Only self-assessments were used. The relatively low number of subjects in a particular sample could explain the negative results due to the insufficient power of the statistical tests. The results of the present study support the notion that trait anhedonia in severe depressives could be a predictor of a high risk of suicide and, therefore, could be screened in these depressives.

## Figures and Tables

**Table 1 ijerph-19-16147-t001:** Characteristics of the three groups (74 severe major depressives (sample 1), 43 depressives (sample 2) and 36 depressives (sample 3)) and their relationship between recent suicidal ideation and socio-demographical or psychometrical variables using the Pearson correlation coefficient (r) or Student’s *t*-test (t).

		N = 74			N = 43			N = 36	
Variables	M (Sd) or N	r or t	*p*	M (Sd) or N	r or t	*p*		r or t	*p*
**Gender (M/F)**	18/56	0.56	ns	8/35	0.34	ns	26/10	−0.18	ns
**Age**	45.6 (12.7)	−0.04	ns	43.14 (18.77)	−0.07	ns	63.72 (15.66)	−0.18	ns
**CA-BDI**	12.43 (3.08)	0.28	<0.05	4.86 (3.18)	0.27	ns	3.31 (2.84)	0.4	<0.05
**PAS-ANT**	4.09 (1.85)	0.4	<0.05				3.03 (1.42)	−0.01	ns
**PAS-CONS**	7.18 (3.18)	0.35	<0.05				6.39 (2.56)	0.18	ns
**TEPS-ANT**	33.09 (9.52)	−0.16	ns						
**TEPS-CONS**	32.28 (8.43)	−0.33	<0.05						
**SHAPS**	4.58 (3.02)	0.3	<0.05	1.28 (1.76)	−0.07	ns	2.36 (2.62)	0.2	ns
**ANH-BDI**	6.03 (1.67)	0.1	ns	3.09 (2.08)	0.37	<0.05	3.75 (1.96)	0.08	ns
**LI-BDI**	2.05 (0.92)	0.04	ns	1.02 (0.94)	0.3	ns	1.03 (0.91)	0.19	ns
**SLIPS**	31.28 (12.43)	0.18	ns						
**MS**	38.81 (12.92)	−0.07	ns	41.53 (11.26)	−0.31	<0.05			
**EE**	45.73 (14.89)	−0.13	ns	46.88 (12.65)	0.02	ns			
**MR**	38 (19)	0.01	ns	41.3 (13.96)	−0.15	ns			
**SM**	36.82 (13.74)	−0.11	ns	46.44 (12.34)	−0.21	ns			
**SR**	42.63 (11.78)	−0.09	ns	46.04 (10.97)	−0.13	ns			
**MuR**	36.18 (15.35)	−0.09	ns	42 (12.9)	−0.2	ns			
**GROUP**	(16 mel vs. 58 nmel)	0.35	ns						
**SID**	1.38 (0.84)			0.42 (0.73)			0.28 (0.45)		

CA-BDI (cognitive–affective symptoms of depression); The Anticipatory (PAS-ANT) and Consummatory subscales (PAS-CONS) of the Physical Anhedonia Scale (PAS); the Anticipatory (TEPS-ANT) and Consummatory subscales (TEPS-CONS) of the Temporal Experience of Pleasure Scale (TEPS); SHAPS (Snaith–Hamilton Pleasure Scale); anhedonia subscale (ANH-BDI) of the Beck Depression Inventory-II (BDI-II); item # 12 of the BDI-II (LI-BDI); the Specific Loss of Interest and Pleasure Scale (SLIPS); Barcelona Music Reward Questionnaire (BMRQ) with five subscales: musical seeking (MS), emotion evocation (EE), mood regulation (MR), sensory–motor (SM), and social reward (SR) and a global score (music reward, MuR); mel (melancholics); nmel (non-melancholics); SID: suicidal ideation.

## Data Availability

Requests for access will be reviewed by the corresponding author.

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
