# Peer review of "Relationships between Recent Suicidal Ideation and Recent, State, Trait and Musical Anhedonias in Depression"

_ijerph, 2022, doi:10.3390/ijerph192316147_

Round 1
Reviewer 1 Report
Authors investigated the correlation between several anhedonia scales and suicidal ideation separately in three samples: 74 outpatients with severe depression recruited from two psychiatric departments; 43 outpatients with mild/moderate depression recruited from general practitioners; 36 inpatients with mild/moderate depression recruited from coronary/cardiology units. The paper needs to be markedly improved especially in the explanation of Methods and presentation of Results.
Major comments
A bivariate analysis was carried out, reporting Pearson correlation coefficient and t test obtained from simple linear regression of scores of suicidal ideation and anhedonia measures. First of all, please clarify for each scale if a higher score corresponds to more/less severe anhedonia. Thereafter, clarify that the t test is the b/SE ratio resulting from simple linear regression. Lastly, since the response variable (suicidal ideation) varies only between 0 and 3, please consider to apply also a non-parametric regression or some other non-parametric analysis, at least as a sensitivity analysis.
I wonder if sample 2 and 3 representing patients with mild/moderate depression can be grouped, even if I acknowledge that the setting of care and circumstances of life are markedly different. Anyway, since results between group 2 and 3 are different, this should be commented. Maybe, due to small numbers and specific setting of care, results from group 3 are more difficult to interpret.
Study results are displayed only in Table 1. I suggest to expand the Table and to add Mean/SE of each variable for each study group. Please clarify why some anhedonia scales are lacking in group 2 or group 3. I suggest also to display and comment the scores of suicidal ideation found in the three groups (maybe with a separate graph).
The discussion is somewhat hard to follow. Maybe the distinction between positive/negative results is not of great help; separate sub-sections focusing on severe and mild/moderate depression might be more interesting. Results from previous papers should be more briefly presented, and grouped in view of study results.
Some comments do not seem to be supported by analyses presented in the manuscript, e.g. page 8, “The results of the present study confirmed the significant relationship between suicidal ideation and depression intensity….”
Minor comments
In the Abstract, original numbers of patients before selection based on the presence of depression do not need to be reported. Instead, the number of depressive subjects in each sample, representing the study population, need to be moved earlier (third sentence of the Abstract).
I’m not a native English speaker, but the manuscript needs to be copy-edited (e.g. page 2, line 85, “two psychiatric departments” instead of “to psychiatric departments”)
Author Response
Responses to the reviewer 1
- For each anhedonia scale (see pages 4-5) the meaning of higher score is precised.
- Test t is the Student t test for comparison when the variable is a category (gender.) and Pearson r correlation coefficient between two quantitative variables.
- Additional mutiple regression taking into account that item 9 is an ordinal variable has been done (see page 7).
- Table 1 was completed with the mean with SD or number of subjects
- Some anhedonia scales are lacking because the present study brought together several studies carried out separately.
- A comment concerning the score on item 9 is given (see page 9)
- The organization of the discussion is changed presenting the results in severe depression and then the results in moderate and mild depression.
- One comment has been changed as the relationship between CA-BDI and suicidal ideation was found in two groups and not three. See page 9.
- The abstract was changed .
Reviewer 2 Report
The paper deals with an interesting relationship between the emotional experiences of anhedonia and suicidal ideations.
The study appears promising, however some weaknesses are seen that prevent the publication of the current study at present:
-The introduction does not adequately define the concepts of anhedonia and the different types, moreover some studies previously carried out on the subject should be more explicit; furthermore, the contribution of the research carried out with respect to the state of the art in the scientific literature is not clear.
-The methodology is not clear enough, the reasons why these particular tasks were selected were not understood, the motivation should be deepened and supported by argued hypotheses, information and details on the procedures are lacking
-The results and discussions are poorly elaborated, closely described the outcome of the data analysis, there is a lack of links between research hypotheses, specific objectives of the study and results.
-The implications for health policies are missing.
Author Response
Responses to the reviewer 2
- In the introduction the concept of anhedonia was presented.
- The rationale for the study is presented clearly in the introduction.
- The organization of the discussion is changed presenting the results in severe depression and then the results in moderate and mild depression.
- The implication of the results is mentionned in the conclusion
Round 2
Reviewer 1 Report
In my opinion the manuscript is suitable for publication, pending minor English editing